# Impact of BCR-ABL1 Transcript Type on Response, Treatment-Free Remission Rate and Survival in Chronic Myeloid Leukemia Patients Treated with Imatinib

**DOI:** 10.3390/jcm10143146

**Published:** 2021-07-16

**Authors:** Sílvia Marcé, Blanca Xicoy, Olga García, Marta Cabezón, Natalia Estrada, Patricia Vélez, Concepción Boqué, Miguel Sagüés, Anna Angona, Raúl Teruel-Montoya, Francisca Ferrer-Marín, Paula Amat, Juan Carlos Hernández-Boluda, Mariana M. Ibarra, Eduardo Anguita, Montserrat Cortés, Andrés Fernández-Ruiz, Sandra Fontanals, Lurdes Zamora

**Affiliations:** 1Myeloid Neoplasms Group, Hematology Department, ICO Badalona-Hospital Germans Trias i Pujol, Josep Carreras Leukaemia Research Institute (IJC), 08916 Badalona, Spain; bxicoy@iconcologia.net (B.X.); olga.garcia.calduch@fundacionpethema.es (O.G.); mcabezon@iconcologia.net (M.C.); nestrada@carrerasresearch.org (N.E.); lzamora@iconcologia.net (L.Z.); 2Hematology Department, ICO Hospitalet-Hospital Duran y Reynals, 08908 l’Hospitalet de Llobregat, Spain; pvelez@iconcologia.net (P.V.); c.boque@iconcologia.net (C.B.); frizandres@gmail.com (A.F.-R.); 3Hematology Department, ICO Girona-Hospital Josep Trueta, 17007 Girona, Spain; msagues@iconcologia.net (M.S.); angona.figueras@iconcologia.net (A.A.); 4Hematology-Oncology Department, Hospital General Universitario Morales Meseguer-CIBERER, UCAM, 30008 Murcia, Spain; raulteruelmontoya@hotmail.com (R.T.-M.); fferrermarin@gmail.com (F.F.-M.); 5Hematology Department, Hospital Clínico Universitario-INCLIVA de Valencia, 46010 Valencia, Spain; paulaamat@yahoo.es (P.A.); hernandez_jca@gva.es (J.C.H.-B.); 6Hematology Department, Hospital Clínico San Carlos (HCSC), Instituto de Medicina de Laboratorio (IML), Instituto de Investigación Sanitaria San Carlos (IdISSC), Department of Medicine, Universidad Complutense de Madrid (UCM), 28040 Madrid, Spain; marianamim@gmail.com (M.M.I.); eduardo.anguita@salud.madrid.org (E.A.); 7Hematology Department, Hospital General de Granollers, 08402 Granollers, Spain; mcortes@fphag.org; 8Pharmaceutics Department, ICO Hospitalet-Hospital Duran y Reynals, 08908 l’Hospitalet de Llobregat, Spain; sfontanals@iconcologia.net

**Keywords:** chronic myeloid leukemia, *BCR**-ABL1* transcripts, response to imatinib, survival, discontinuation, relapse-free survival

## Abstract

The most frequent *BCR-ABL1*-p210 transcripts in chronic myeloid leukemia (CML) are e14a2 and e13a2. Imatinib (IM) is the most common first-line tyrosine–kinase inhibitor (TKI) used to treat CML. Some studies suggest that *BCR-ABL1* transcript types confer different responses to IM. The objective of this study was to correlate the expression of e14a2 or e13a2 to clinical characteristics, cumulative cytogenetic and molecular responses to IM, acquisition of deep molecular response (DMR) and its duration (sDMR), progression rate (CIP), overall survival (OS), and treatment-free remission (TFR) rate. We studied 202 CML patients, 76 expressing the e13a2 and 126 the e14a2, and correlated the differential transcript expression with the above-mentioned parameters. There were no differences in the cumulative incidence of cytogenetic responses nor in the acquisition of DMR and sDMR between the two groups, but the e14a2 transcript had a positive impact on molecular response during the first 6 months, whereas the e13a2 was associated with improved long-term OS. No correlation was observed between the transcript type and TFR rate.

## 1. Introduction

Chronic myeloid leukemia (CML) is characterized by a reciprocal translocation between chromosomes 9 and 22, known as the Philadelphia chromosome (Ph), containing the *BCR-ABL1* fusion gene. The most frequent breakpoint of chromosome 22, known as the major breakpoint cluster region (M-bcr), comprises exon 13 or exon 14, generating the e13a2 (b2a2) or the e14a2 (b3a2) transcripts. Both are translated into a constitutively active protein of 210 kDa that differs in 25 amino acids more from the e14a2 transcript [1,2,3].

Imatinib (IM) is a tyrosine–kinase inhibitor (TKI) that inhibits the BCR-ABL1 protein in Ph-positive CML [4]. IM is the most widely used TKI for first-line therapy in CML patients, with most patients achieving optimal hematologic, cytogenetic, and major (MR^3.0^) and deep molecular responses (MR^4.0^ or better) [5]. However, IM fails in some patients and very few of them progress to acute leukemia. The causes of resistance to IM are heterogeneous and have been investigated in depth. On the other hand, in some non-resistant patients, minimal residual disease persists during TKI therapy. One hypothesis is that the difference in M-*bcr* breakpoint may be a cause of this persistence [3]. Some studies carried out prior to the IM era did not find any influence of *BCR-ABL1* transcript type on the clinical outcome [6,7,8]. However, in the IM era, a Brazilian study, including a small number of patients at different phases of the disease, suggested that patients with the e13a2 transcript were more sensitive to IM in terms of molecular response [9], whereas other studies with larger number of patients demonstrated a better molecular response to IM in patients with the e14a2 transcript [3,10,11,12].

The achievement of deep molecular response (DMR) and its duration (sDMR) for at least 2 years are the current goal for discontinuing TKI treatment in CML. Approximately 50% of patients with sDMR can remain relapse-free after discontinuing TKI treatment in the so called treatment-free remission (TFR) phase, defined as the time from TKI discontinuation to the date of restarting therapy or the date of last control if treatment is not restarted. Therefore, there is a need to identify factors at diagnosis that will be associated with the acquisition of DMR, sDMR, and TFR. In this respect, a negative effect of the e13a2 transcript on the probability of achievement of TFR has been reported [13,14,15].

The aim of the present study was to evaluate the influence of the e13a2 and e14a2 *BCR-ABL1* transcripts on outcomes of IM treatment in a multicenter series of 202 CML patients.

## 2. Material and Methods

### 2.1. Patient Samples

We selected patients from databases from seven Spanish centers diagnosed of chronic phase CML from 1999 to 2016 and treated with IM at first-line according to the local standard of care and with a minimum follow-up of 18 months in the majority of cases. Samples were collected after obtaining informed consent in accordance with approved Institutional Research Boards protocols and the Declaration of Helsinki (Ref. CEI: PI-15-007, Code: ICO-ITK-2015-01 (ISOF-p210-LMC)).

Peripheral blood (PB) and bone marrow (BM) (when applicable) were obtained at diagnosis and after 3, 6, and 12 months of IM treatment and during the follow-up. The main demographic and clinical characteristics were collected from the records of all patients. Cytogenetic and molecular responses were analyzed according to the 2013 European Leukemia Net (ELN) guidelines [16].

### 2.2. Cytogenetic Studies

Conventional chromosome G-banding (CG-banding) was performed at diagnosis and at 3, 6, and 12 months. The analyses at 6 and 12 months were performed in patients who did not achieve optimal cytogenetic response at 3 and 6 months, respectively.

Fluorescent “in situ” hybridization (FISH) at diagnosis was performed in cases where CG-banding was not possible, according to the manufacturer’s instructions using LSI BCR/ABL dual-color, dual-fusion (Vysis-Abbott Molecular). At least 200 interphase nuclei were analyzed from each case.

### 2.3. Determination of BCR-ABL1 Transcript Type

Whole PB or BM samples were collected in 10 mL or 3 mL EDTA tubes, respectively. RNA was isolated from bone marrow or peripheral blood total leukocytes using TRIzol^®^ reagent (Invitrogen; Carslbad, CA, USA) according to the manufacturer’s protocol. RNA concentration was quantified, and 1 μg of total RNA was reverse transcribed to cDNA using SuperScript IV or MMLV Reverse Transcriptase with random hexamers/primers according to the manufacturer’s protocol (Invitrogen; Thermo Fisher Scientific, Inc, Waltham, MA, USA). *BCR-ABL1* was amplified using PCR primers as previously described [17] using 3 μL of cDNA. PCR products were transferred to a QIAxcel (QIAGEN Inc, Hilden, Germany) to identify the type of transcript expressed depending on its size.

### 2.4. Real-Time Quantitative PCR

Real-time quantitative PCR was carried out in an ABI7900 PCR thermal cycler (Applied Biosystems, Foster City, CA, USA). Messenger RNA (mRNA) expression of *BCR-ABL1* was calculated as a percentage relative to *ABL1* expression and adjusted to the international scale (IS). The ratio was calculated at diagnosis and at 3, 6, and 12 months and at different points during follow-up of IM treatment to verify the acquisition of molecular response and deep molecular response.

### 2.5. Statistical Analysis

Baseline characteristics were described as frequency and percentage for categorical variables and median and range for quantitative variables. Comparisons of categorical variables between groups were calculated using the Chi-square or Fisher exact test, when necessary, while the median test was used to compare continuous variables.

Cumulative incidence of cytogenetic and molecular response was analyzed taking into account competing risks. Achieving a response to IM during the first 12 months after IM onset was considered, as main event and time was defined as months from IM onset to response. Patients who died or changed TKI during the first 12 months without achieving response were considered as competing events [4], and time was defined as months from IM onset to date of death or date of IM stop. Patients alive or dead after 12 months without response were considered as censures, and time was defined as months from IM onset to date of last follow-up. The DMR was defined as the time from diagnosis until the time of achievement of MR^4.0^ or MR^4.5^ and was analyzed by competing risks.

OS was defined as time from diagnosis to the last follow-up or death from any cause. Survival probabilities were calculated using the Kaplan–Meier method, and the log-rank test was used for comparisons between groups.

The CIP to acute leukemia was defined as the time from diagnosis to progression or the last follow-up and was analyzed by competing risks, considering patients alive without progression as censures, patients who progressed as events, and patients who died without progression as competing events. 

Both cumulative incidence curves for response and CIP were plotted, and comparison analyses between groups were performed by Gray test.

The relationship between TKI discontinuation and isoform type was studied by logistic regression.

Treatment-free remission (TFR) was defined as time from IM discontinuation to the occurrence of loss of MMR, restart of TKI treatment, progression, or death from any cause, the earliest of these events. TFR probabilities were calculated using the Kaplan–Meier method, and a log-rank test was used for comparisons between groups.

Two-sided *p* values < 0.05 were considered as statistically significant. The statistical package SPSS, version 24.0 (SPSS Inc., Chicago, IL, USA), and R software (version 3.3.2) were used for all analyses.

## 3. Results

### 3.1. Patients

We studied 202 CML patients from seven Spanish centers diagnosed from 1999 to 2016, most of them with a minimum follow-up of 18 months on IM treatment (median 6 years; 0.3–11.6). The molecular analysis at diagnosis showed that 76 (37.6%) patients expressed the e13a2 *BCR-ABL1*-p210 transcript, while 126 (62.4%) expressed the e14a2 transcript. All patients were treated with IM at standard doses as first-line TKI therapy.

The demographic and clinical characteristics were not significantly differing between the two groups. Patient distribution according to Sokal [18], EUTOS [19], and ELTS [20] scores was also comparable (Table 1).

Some patients showed suboptimal response or were intolerant to IM (Table 2). Nineteen patients required an increase in IM dose (five with e13a2 and 14 with e14a2), while a change to a second-line treatment was needed in 96 patients (37 with e13a2 and 59 with e14a2). Most of these patients switched beyond 12 months of treatment, and patients that did so earlier were intolerant. The reason for switching to a second-generation TKI was not influenced by the transcript types (Table 2).

### 3.2. Cytogenetics According to the Transcript Type

Cytogenetic studies were available in 182 patients at diagnosis. In 154 out of 182 (84.6%) patients, the t(9;22) was the only alteration found. Additional chromosomal abnormalities did not differ between the two groups (Table 1).

We analyzed the cumulative incidence of cytogenetic response during the first 12 months of treatment and differences in the two groups regarding major and complete cytogenetic response (MCyR and CCyR) were comparable (Figure 1A).

### 3.3. Molecular Response by Transcript Type

The number of patients with molecular response during follow-up was similar in both groups (Figure 1B), but a trend of better cumulative incidence of MR^3.0^ was observed at 6 months in patients expressing e14a2 (Figure 1C).

### 3.4. Effect of Transcript Type on Deep Molecular Response (DMR)

One hundred and four out of 202 patients (51.4%) who started IM as first-line obtained a DMR within two years. No statistically significant differences were observed in the cumulative incidence of DMR 4.0 and 4.5, respectively, at 2 years between the e13a2 and e14a2 groups (Figure 2A,B). On the other hand, a better DMR (4.0 and 4.5) in patients with a low and intermediate Sokal score as compared to patients with a high Sokal score was observed (33% [23%, 43%] vs. 22% [15%, 30%], *p* = 0.0169 and 27% [17%, 37%] vs. 14% [8%, 21%], *p* = 0.018, respectively) (Figure 2C,D).

### 3.5. Prognostic Impact of Transcript Type on Overall Survival and CIP

The 8-year OS probability (95% CI) of the whole cohort was 87% (82–92%) (Figure 3A), being significantly better in the e13a2 group (Figure 3B). At the standard IM dose, 25% and 22% of patients in the e13a2 and e14a2 groups, respectively, did not achieved the optimal response to treatment during follow-up. Eight of these patients (two in the e13a2 group and six in the e14a2 group) progressed to accelerated phase without significant differences in the 8-year CIP between the two groups (Figure 3C), and with one out of two (50%) and four out of six (67%), respectively, dying after progression to blast phase due to CML causes (two patients died due to cerebral hemorrhage, two due to infection, and one due to multi-organ failure).

### 3.6. Analysis of TKI Discontinuation and Treatment-Free Remission (TFR) According to the Transcript Type

Data of TKI discontinuation and TFR were available in 104 patients that acquired DMR. Only 66 of these patients achieved sDMR (25 with the e13a2 and 41 with the e14a2 transcript) during at least two years, and 36 of them discontinued IM treatment after a minimum of five years of treatment (Appendix A). Within this group, 16 (44%) had the e13a2 transcript and 20 (56%) the e14a2. There was no correlation between the transcript type and the fulfillment of discontinuation criteria (16/24 e13a2 (67%) and 20/36 e14a2 (56%), *p* = 0.389). 

TFR (95% CI) of the whole series of discontinued patients (*n* = 44, 36 with sDMR and eight without sDMR due to toxicity or patient’s decision) was 63% (48%, 78%) (Figure 4A). Fifteen out of 44 cases relapsed, being the median follow-up after discontinuation of non-relapsed patients of 28 months (range 0, 65), but all of them regained molecular response when reintroducing the TKI. Differences in the TFR between e14a2 and e13a2 transcript were not statistically significant (55% [32%, 78%] vs. 69% [49%, 89%], respectively, *p* = 0.265) (Figure 4B).

Duration of IM treatment and DMR duration before discontinuation did not significantly correlate with TFR in our cohort (HR [95% CI]: 0.99 [0.98, 1.01], *p* = 0.237 and 0.99 [0.98, 1.01], *p* = 0.372, respectively).

## 4. Discussion and Conclusions

This multicenter study performed in 202 patients with chronic phase CML treated with IM as first-line therapy showed a higher percentage of major molecular response at 6 months in the e14a2 group that also presented a descriptively tendency to better TFR rates, even though no differences in the cumulative incidence of cytogenetic and molecular responses nor in the acquisition of DMR and sDMR between the two groups were found. Although transcript type had no impact on CIP, patients with the e13a2 transcript had better OS.

The majority of CML patients have a *BCR-ABL1*-p210 with an e13a2 or an e14a2 transcript [1,21]. Previous studies have shown that the e13a2 transcript is more frequent in young men and is associated with a blast crisis of myeloid phenotype [8,12,22,23]. The e14a2 transcript has been associated with higher platelet count [12,24], while its relationship with lower white cell counts is controversial [11,12,25]. In our series, we did not observe any of these correlations, not even with sex, age, splenomegaly, or platelet and blast count. Patients with CML were stratified in low, intermediate, or high-risk groups according to EUTOS or Sokal scores [18,19]. The recently developed ELTS score divides the patients in three different risk groups according to the probability of dying of CML improving prognostication compared to EUTOS and Sokal scores [20]. Nevertheless, patient distribution according to Sokal, EUTOS, and ELTS scores between the two groups was comparable in our study so that transcript type did not affect the patients’ risk stratification.

We did not find a correlation between the presences of additional chromosomal abnormalities at diagnosis in the two groups. Intolerance, suboptimal response, or failure to achieve optimal response to IM were also similar in both groups. Our results are in accordance with previous studies that did not find any correlation between the transcript type and clinical characteristics and prognostic scores [26,27,28].

In the pre-IM era, no differences in outcomes were found between patients with CP-CML expressing the e13a2 or e14a2 transcripts [6,7,8,24,29]. In the IM era, some studies have addressed the influence of the *BCR-ABL1*-p210 transcript type with response to IM treatment and CML outcome with controversial results [9,10,11,12]. Lucas et al. [2] found that the one-year CCyR rate in 78 patients treated with IM 400 mg daily was higher in e14a2 patients than in those with e13a2. In a larger study, Hanfstein et al. [25] demonstrated that the probability of achieving MR^3.0^ or MR^4.0^ was significantly higher in e14a2 than in e13a2 patients. Baccarani et al. [3] described no statistically significant differences in CCyR achievement, while the major molecular response rate was lower in e13a2 patients in seven of the eight studies analyzed. In our analysis, the cumulative incidence of cytogenetic and molecular responses during the first 12 months of IM treatment was similar in both groups of patients, with optimal responses at 3 and 6 months being also similar. Nevertheless, we observed a better cumulative incidence of major molecular responses at 6 months in the e14a2 group, comparable to previously reported data, and supporting the concept that this transcript type may be more sensitive to IM [3,11,12,30,31].

Correlation between *BCR-ABL1* transcript type and the probability of achieving DMR and its duration has less frequently been investigated. However, some studies demonstrated a significantly lower rate of DMR in the e13a2 subgroup [3,14,23], while few studies showed that patients with the e14a2 transcript have prolonged sDMR [14,15]. In our study, we did not observe differences in the number of patients that acquired DMR between the two groups nor in the time of the achievement such DMR and the sDMR, maybe because some patients changed to a second TKI and were censored for the analysis at that time. On the contrary, we found a tendency and a significant correlation between ELTS and Sokal scores, respectively, and the acquisition of DMR according to what was published by Breccia et al. [15] who found an association between a stable DMR and Sokal risk in univariate and multivariate analysis.

It has also been suggested that the transcript type not only affected response outcomes, but also long-term OS and CIP to acute leukemia [11]. Additionally, the e14a2 transcript has been correlated with longer OS, and on the contrary, patients with e13a2 may have an aggressive course. Although previous studies did not show any difference in event-free survival (EFS) or transformation-free survival (TFS) according to transcript type among patients treated with IM [22,32], Jain et al. [11] analyzed the impact of transcript types in 481 patients with CML treated with different TKI regimens and suggested that the e13a2 transcript may negatively impact EFS and TFS, but these results were not statistically significant. Castagnetti et al. [4] demonstrated a better 7-year OS in patients with the e14a2 transcript. In contrast, Baccarani et al. [3] did not observe differences in the impact of transcript type in PFS and OS, and Pagnano et al. [32] observed a superior 10-year OS in patients with e13a2 transcript, although they could not find differences at 5-year OS. In our study, we found significant differences in OS between the two groups, with the e13a2 transcript having a better OS than the e14a2 transcript despite a tendency to better molecular response at 6 months in the e14a2 group. One plausible explanation of this discrepancy might be that due to the efficacy of TKI treatment most patients die in the long term from causes other than CML due to older age, toxicities of TKIs, or co-morbidities especially under treatment with second- or third-generation TKI. Other factors such as the resistance to more than one TKI might also play a role in the long-term OS. All these data suggest several discrepancies between studies about OS and transcript type relationship that should be investigated in larger trials [12]. Patients with e14a2 transcript predicted for longer transformation-free survival and lower progression to acute leukemia [11]. We did not observe differences in CIP between the two groups with only eight patients who progressed to accelerated phase and five out of eight who died due to CML. Although these patients who died were young patients with high-risk Sokal and ELTS scores, and nowadays, they would be treated with second-generation TKI, they were treated with imatinib because no other options as first line were possible at the moment of their diagnosis.

Some limitations of our study are the limited number of cases available for analyzing the endpoints of our study, its retrospective and multicenter nature, and the different treatment approaches that have emerged throughout the years since IM became available. There may be also technical limitations, because the same PCR assay is used to amplify both transcripts variants, and some studies have suggested that e14a2 amplification may be technically limited during RT-qPCR quantification [33,34]. Moreover, a polymorphism in exon 13 of *BCR* that reduced the efficiency of the primers used to amplify e13a2 has been described [3]. 

There are few studies focused on the effect of transcript type and the maintenance of a long-term TFR after TKI discontinuation. D’Adda et al. [14] reported the negative effect of e13a2 transcript on a durable TFR in a larger series of 173 patients and suggested that differences in immunogenic ability between the two transcript types may be associated with the risk of relapse after TKI discontinuation [14], and other studies postulated that IM treatment and DMR duration may modulate the immunogenic system [35,36]. Our results, with a lower number of cases, are in line with D’Adda et al.’s study suggesting a negative effect of e13a2 transcript on TFR.

In summary, in this CP-CML series of patients treated with IM in first line, the e14a2 transcript type may have a positive impact on cumulative incidence of major molecular response at 6 months, whereas the e13a2 transcript type might be associated with improved long-term OS. The influence of transcript type on outcomes in CML remains controversial and needs further investigations with a larger series to consider transcript type as a prognostic factor associated with TFR. For the time being, we think that the transcript type (e13a2 vs. e14a2) should not guide treatment decision making in CML.

## Figures and Tables

**Figure 1 jcm-10-03146-f001:**
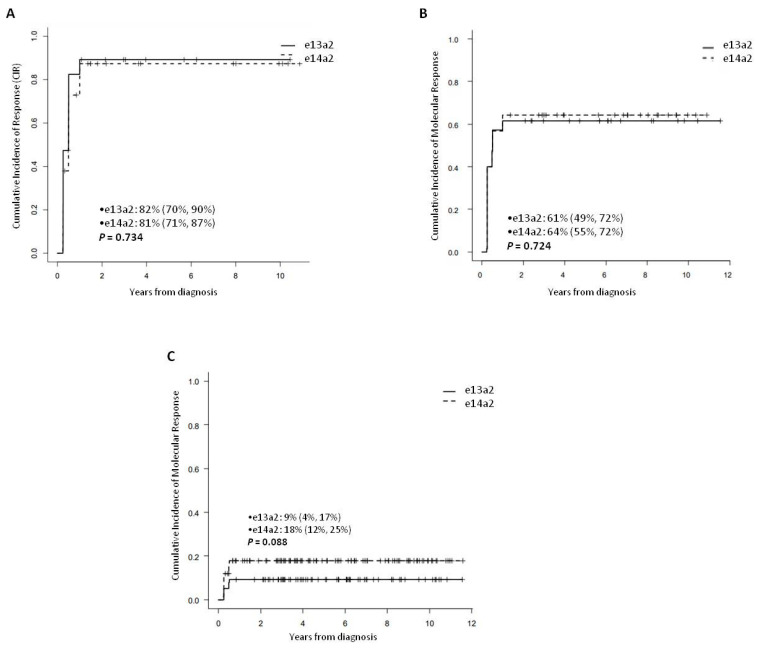
Cumulative incidence of cytogenetic response at 12 months (**A**) and molecular response MR^3.0^ at 12 months (**B**) and at 6 months (**C**) according to the transcript type in our series of patients.

**Figure 2 jcm-10-03146-f002:**
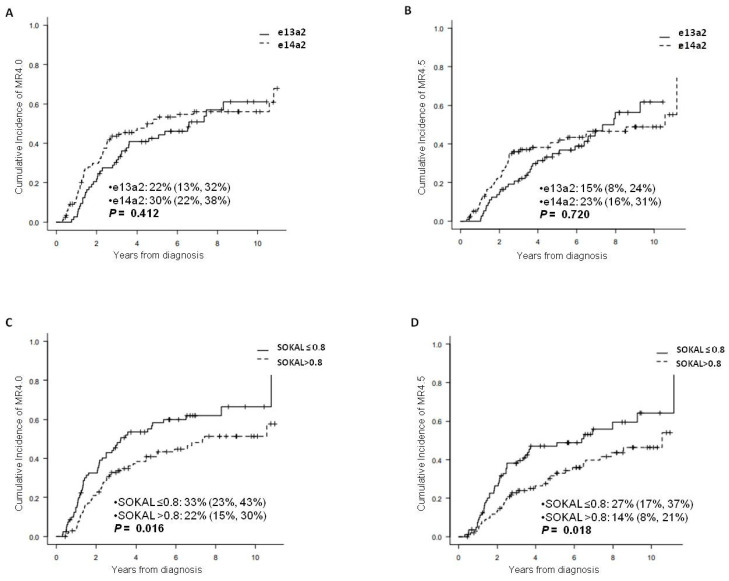
Cumulative incidence of deep molecular response 4.0 and 4.5 according to the transcript type ((**A**,**B**)) and according to Sokal score ((**C**,**D**)) at 2 years, respectively.

**Figure 3 jcm-10-03146-f003:**
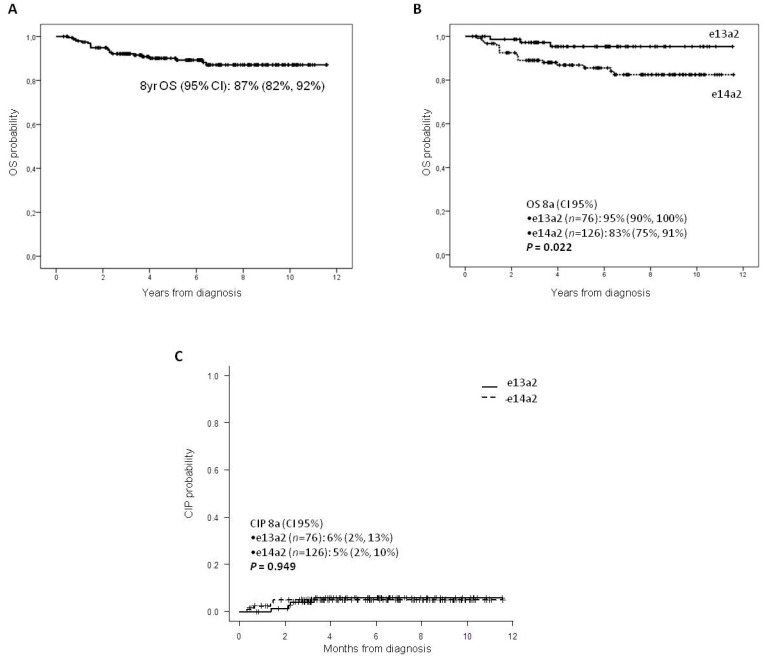
Overall survival in the whole series (**A**) and according to the transcript type (**B**). Cumulative incidence of progression (CIP) according to the transcript type in our series of patients (**C**).

**Figure 4 jcm-10-03146-f004:**
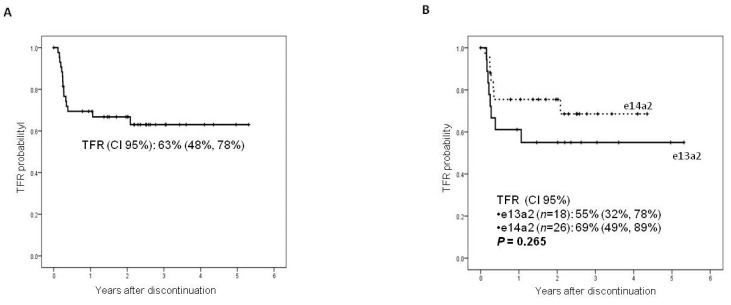
Analysis of treatment-free remission (TFR) in the global series (**A**) and in the two groups according to the transcript type (**B**).

**Table 1 jcm-10-03146-t001:** Demographic and baseline clinical characteristics of the study series.

		e13a2 (*n* = 76)	e14a2 (*n* = 126)	Total (*n* = 202)	*p* Value
Sex (male/total) (%)		37/76 (49)	67/127 (53)	104/202 (52)	0.536
Age, median (min, max)		54 (15, 83)	57 (12, 84)	56 (12, 84)	0.828
Splenomegaly, *n* (%) Median cm (min, max)		23/75 (31) 4 (2, 20)	39/120 (33) 5 (1, 30)	62/195 (32) 5 (1, 30)	0.789 0.542
Platelets (x10e9/L), median (min, max)		314 (37, 2236)	386 (21, 1872)	364 (21, 2236)	0.241
Blasts PB (%), median (min, max)		0 (0, 11)	0 (0, 11)	0 (0, 11)	0.918
Cytogenetics, *n* (%)	t(9;22)	61/68 (90)	99/116 (85)	160/184 (87)	0.397
	t(9;22) + others	7/68 (10)	17/116 (15)	24/184 (13)	
EUTOS, median (min, max)		28 (0, 143)	28 (0, 148)	28 (0, 148)	0.358
EUTOS > 87, *n* (%)		3/75 (4)	13/120 (11)	16/195 (8)	0.091
Sokal, n (%)	Low risk	39/75 (52)	46/120 (38)	85/195 (44)	0.173
	Int risk	26/75 (35)	54/120 (45)	80/195 (41)	
	High risk	10/75 (13)	20/120 (17)	30/195 (15)	
ELTS, n (%)	Low risk	43/75 (57)	61/120 (51)	104/195 (53)	0.376
	Int/High risk	32/75 (43)	59/120 (49)	91/195 (47)	

PB: peripheral blood; ELTS: EUTOS long-term survival score; Int: intermediate.

**Table 2 jcm-10-03146-t002:** Intolerance, suboptimal response, and failure to imatinib treatment according to the transcript type.

	e13a2 (*n* = 76)	e14a2 (*n* = 126)	Total (*n* = 202)	*p* Value
Intolerance, *n* (%)	12/76 (16)	22/126 (18)	34/202 (17)	0.758
Suboptimal Response, *n* (%)	13/76 (17)	19/126 (15)	32/202 (16)	0.702
Failure, *n* (%)	19/76 (25)	28/126 (22)	47/202 (23)	0.651

## Data Availability

The data present in the study can be partialy available upon the request to corresponding author, under national regulations for data sharing.

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
