# Peer review of "Impact of BCR-ABL1 Transcript Type on Response, Treatment-Free Remission Rate and Survival in Chronic Myeloid Leukemia Patients Treated with Imatinib"

_jcm, 2021, doi:10.3390/jcm10143146_

Round 1

Reviewer 1 Report

Marcè and Colleagues presented a study focused on the evaluation of the impact of BCR-ABL1 transcript type on different clinical outcomes. The study is not so innovative and conclusive. The data are partially confirming and partially confuting results of other studies previously published. I have some questions and some comments in order to better appreciate the manuscript.

  • The statistical analysis paragraph is quite confusing. In particular the sentence from line 114 to line 118. Please, rephrase it. Moreover, no calculation of the sample size has been reported. How was 202 patients calculated. In the same paragraph, line 119 "RM". What do the Authors refer to? Is it Molecular Response "MR"? Please, specify it.
  • In the paragraph 2.3 please describe the sample manipulation (e.g. centrifuge) and the RNA extraction and retro-transcription. It is mandatory, in particular because it is a multicenter study. 
  • In Table 1, all the measure units must be reported, in particular for platelets and blasts count.
  • In paragraph 3.4, please report all the p of significance also in the text in order to help the reader.
  • In paragraph 3.5, all the cause of death must be reported. If the Authors agree, even a table may be added.
  • In the Discussion section, additional aspects must be considered. E.g. Concerning the technical limits of RT-qPCR, both the results presented by the Danish and the Italian Group (PMID 30985947 and 31233644) suggest that RT-qPCR quantification may influence the analysis of the transcripts types impact. In fact, it seems that e14a2 amplification is technically limited during RT-qPCR quantification, but well performed by other tools, such as digital PCR. Please add a comment on both the manuscripts.

Author Response

Response to Reviewer 1 Comments:

Point 1. The statistical analysis paragraph is quite confusing. In particular the sentence from line 114 to line 118. Please, rephrase it. Moreover, no calculation of the sample size has been reported. How was 202 patients calculated. In the same paragraph, line 119 "RM". What do the Authors refer to? Is it Molecular Response "MR"? Please, specify it.

Response 1. We have re-written the statisitical analysis paragraph trying to make it clearer (page 3, line 123, paragragraph 2.5: Cumulative incidence of cytogenetic and molecular response was analyzed taking into account competing risks. Achieving response to IM during the first 12 months after IM onset was considered as main event and time was defined as months from IM onset to response. Patients who died or changed TKI during the first 12 months without achieving response were considered as competing events [4] and time was defined as months from IM onset to date of death or date of IM stop. Patients alive or dead after 12 months without response were considered as censures and time was defined as months from IM onset to date of last follow-up).

As it is a retrospective study, patients who fulfilled the inclusion criteria were selected from databases from 7 institutions from the Spanish Group of CML that agreed to participate. We have modified the sentence to make it clearer (page 2, line 78, paragraph 2.1: We selected patients from databases from seven Spanish centers diagnosed of chronic phase CML from 1999 to 2016 and treated with IM at first-line according to the local standard of care and with a minimum follow-up of 18 months in the majority of cases).

We have corrected the mistake and we have changed RM to MR as it means molecular response (page 3, line 129: ... achievement of MR4.0 or MR4.5)

Point 2. In the paragraph 2.3 please describe the sample manipulation (e.g. centrifuge) and the RNA extraction and retro-transcription. It is mandatory, in particular because it is a multicenter study.

Response 2. We have included the RNA extraction and retro-transcription methods used in the different centers in the paragraph 2.3 (page 3, lines 99: Whole PB or BM samples were collected in 10 mL or 3 mL EDTA tubes, respectively. RNA was isolated from bone marrow or peripheral blood total leukocytes using TRIzol® reagent (Invitrogen) according to the manufacturer’s protocol. RNA concentration was quantified and 1 μg of total RNA was reverse transcribed to cDNA using SuperScript IV or MMLV Reverse Transcriptase with random hexamers/primers according to the manufacturer’s protocol (Invitrogen; Thermo Fisher Scientific, Inc.). BCR-ABL1 was amplified using PCR primers as previously described [17] using 3 μl of cDNA. PCR products were transferred to a QIAxcel (QIAGEN Inc, USA) to identify the type of transcript expressed depending on its size).

Point 3. In Table 1, all the measure units must be reported, in particular for platelets and blasts count.

Answer 3. We have included the measure units in the table 1 (see table 1 in page 4).

Point 4. In paragraph 3.4, please report all the p of significance also in the text in order to help the reader.

Response 4. We have introduced the “p” value in the text of paragraph 3.4 (page 6, line 189: … Sokal score was observed (33% [23%, 43%] vs. 22% [15%, 30%], p=0.0169 and 27% [17%, 37%] vs. 14% [8%, 21%], p=0.018, respectively) (Figure 2C and D)).

Point 5. In paragraph 3.5, all the causes of death must be reported. If the Authors agree, even a table may be added.

Response 5. We have included the causes of death of the 5 patients in the text of paragraph 3.5 (page 6, line 202: … dying after progression to blast phase due to CML causes (2 patients died due to cerebral hemorrhage, 2 due to infection and 1 due to multi-organ failure)).

Point 6. In the Discussion section, additional aspects must be considered. E.g. Concerning the technical limits of RT-qPCR, both the results presented by the Danish and the Italian Group (PMID 30985947 and 31233644) suggest that RT-qPCR quantification may influence the analysis of the transcripts types impact. In fact, it seems that e14a2 amplification is technically limited during RT-qPCR quantification, but well performed by other tools, such as digital PCR. Please add a comment on both the manuscripts.

Response 6. These two articles mentioned the discrepancy observed with RT-qPCR quantification between the two transcripts. Kjaer et al. not only could explain the reason for this discrepancy but also recognized an average fold discrepancy between centers. Our analysis eas restricted to validated and standardized method RT-qPCR, which is widely used. On the contrary, digital PCR is a technique that needs validation and standardization and is not available in most centers. Even though we have added a sentence and the two references to consider reviewer’s comment (page 11, line 315: … same PCR assay is used to amplify both transcripts variants and some studies have suggested that e14a2 amplification may be technically limited during RT-qPCR quantification [33,34]; Pages 15, lines 440-445: 33. Kjaer, L.; Skov, V.; Andersen, M.T.; Aggerholm, A.; Clair, P.; Gniot, M.; Soeby, K.; Udby, L.; Dorff, M.H.; Hasselbalch, H.; et al. Variant-specific discrepancy when quantitating BCR-ABL1 e13a2 and e14a2 transcripts using the Europe Against Cancer qPCR assay. Eur. J. Haematol. 2019, 103(1), 26-34. 34. Bernardi, S.; Bonifacio, M.; Iurlo, A.; Zanaglio, C.; Tiribelli, M.; Binotto, G.; Abruzzese, E.; Russo, D. Variant-specific discrepancy when quantitating BCR-ABL1 e13a2 and e14a2 transcripts using the Europe Against Cancer qPCR assay. Is dPCR the key?. Eur. J. Haematol. 2019, 103(3), 272-273.).

Reviewer 2 Report

Tyrosine kinases inhibitors revolutionized chronic myeloid leukemia treatment for many years. Now, these inhibitors could prolong patients' life expectancy to be comparable to age-matched healthy individuals. Still some patients have variant responses to TKI treatment. So, the authors try to investigate whether the different isoforms of BCR-ABL1 could cause the response differently. Overall, there is a marginal difference between the two isoforms, according to these data, the authors claim that “we think that the transcript type (e13a2 vs e14a2) should not guide treatment decision making in CML.”. The data are convincing, and the presentation and discussion are performed well. I have two major issues: one is the sample size, since it is small, the authors should discuss the limitation more; the other one is the figures, they should be optimized. Other minor issues are listed below:

  1. Some labels are too small in Figure 2.
  2. No figure legend for Figure S1.

Author Response

Response to Reviewer 2 Comments:

Point 1. I have two major issues: one is the sample size, since it is small, the authors should discuss the limitation more.

Response 1. We have emphasized the limitation of the sample size in different paragraphs of the discussion (Page 11, line 310: Some limitations of our study are the limited number of cases available for analyzing the endpoints of our study …; Page 11, line 331: … needs further investigations with larger series to consider transcript type as a prognostic factor ).

Point 2. The other one is the figures, they should be optimized.

Response 2. We have re-edited figures to get them optimized (pages 5, 6, 7 and 8).

Point 3. Other minor issues are listed below:

  1. Some labels are too small in Figure 2.

Response 3.1. We have changed the letter size to visualize it properly (page 6, figure 2).

  1. No figure legend for Figure S1.

Response 3.2. We have written the legend of the figure S1 (manuscript-supplementary).

Round 2

Reviewer 1 Report

The Authors answered to all my questions and replayed to my comments. The manuscript quality has been improved in the revised version and the data are more complete.